# Adult Stem Cell Functioning in the Tumor Micro-Environment

**DOI:** 10.3390/ijms20102566

**Published:** 2019-05-25

**Authors:** Yuhan Jiang, Alan Wells, Kyle Sylakowski, Amanda M. Clark, Bo Ma

**Affiliations:** 1Department of Pathology, University of Pittsburgh, Pittsburgh, PA 15261, USA; yuhan@pitt.edu (Y.J.); kys13@pitt.edu (K.S.); amc235@pitt.edu (A.M.C.); 2Department of Bioengineering, University of Pittsburgh, Pittsburgh, PA 15260, USA; 3Department of Computational & Systems Biology, University of Pittsburgh, Pittsburgh, PA 15260, USA; 4UPMC Hillman Cancer Center, University of Pittsburgh, Pittsburgh, PA 15213, USA; 5VA Pittsburgh Healthcare System, Pittsburgh, PA 15213, USA; 6School of Medicine, Tsinghua University, Beijing 100084, China

**Keywords:** cancer-associated epithelial-to-mesenchymal transition, matricellular proteins, secretome

## Abstract

Tumor progression from an expanded cell population in a primary location to disseminated lethal growths subverts attempts at cures. It has become evident that these steps are driven in a large part by cancer cell-extrinsic signaling from the tumor microenvironment (TME), one cellular component of which is becoming more appreciated for potential modulation of the cancer cells directly and the TME globally. That cell is a heterogenous population referred to as adult mesenchymal stem cells/multipotent stromal cells (MSCs). Herein, we review emerging evidence as to how these cells, both from distant sources, mainly the bone marrow, or local resident cells, can impact the progression of solid tumors. These nascent investigations raise more questions than they answer but paint a picture of an orchestrated web of signals and interactions that can be modulated to impact tumor progression.

## 1. Introduction

Metastases, or dissemination of cancer cells to distant organs with subsequent growth of these cells, are responsible for the majority of deaths due to solid tumors. Upon dissemination, tumor cells step through two situations [1,2,3]. First, the cells must adapt to their new, hostile ectopic environment by undergoing a phenotypic shift that can lead to an extended period of dormancy, which is able to last for years to even decades; during this time the cells are resistant to death signals and chemotherapies and are invisible to the immune system [4]. During the second stage, these cryptic micrometastases emerge and outgrow as aggressive and lethal metastases. Unfortunately, these enlarging tumors acquire new modes of generalized resistance to killing, even if they do now express targets of newer immunomodulatory therapies [4]. To eliminate the mortality related to metastases, we need to keep the early metastases dormant, re-sensitize these growths to therapies, or develop new approaches. Hence, we first need to better understand the cellular behavior and molecular events that enable the dormancy, emergence, and resistances. 

The metastatic nodules appear to be phenotypically plastic without the widespread specific genetic mutations that characterize the initial carcinogenesis [2]. Similarly, generalized resistance of these disseminated cells appear to be imprinted by the context of the cancer cells in the organ, as dormancy and chemoresistance can be reversed by tumor-cell extrinsic signals [5,6]. Thus, we need to explore the localized micro-environment rather than the cancer cells per se.

This specialized ‘organ’ construct, known as the tumor microenvironment (TME), is the tissue space comprised of the cancer cells interacting with surrounding endogenous cells, including parenchymal cells, blood vessels, fibroblasts, matrix, and tissue and hematopoietic immune cells [7,8,9]. Recently, a role for stem cells, both resident in the tissue and recruited from circulation, in regulation of the TME has been proposed. These mesenchymal stem cells (MSCs), also known as multipotent stromal cells, reside in all tissues in addition to circulating from the bone marrow [10,11,12]. MSCs are renowned for their multi-faceted therapeutic potential in tissue repair and wound healing. These cells help reestablish homeostasis not just via expansion and differentiation to provide for cell replacement, but also by altering the resident cells through numerous paracrine signaling cascades, including immune suppressive cytokines, pro-regenerative growth factors, and secretion of extracellular vesicles [13]. It is these signals that may be the greatest effect, as they have progressed to clinical use as pro-regenerative and immunosuppressive therapies while the cellular replacement implementations have lagged [14,15]. In the present review, we discuss these aspects of MSCs, and how the production of these numerous signals may impact the disseminated cells. This review aims to highlight the role of MSCs in tumor progression, the driving of metastasis in particular, in order to provide advanced and comprehensive information on the interaction of MSCs and tumor cells in both primary and metastatic sites.

## 2. Characterization of Human Mesenchymal Stem Cells/Multipotent Stromal Cells (MSCs)

Mesenchymal stem cells/multipotent stromal cells (MSCs) are a subset of non-hematopoietic adult stem cells found in various tissues in the body [12,16]. They serve as the resident tissue sources for precursor cells to aid in tissue replacement and repair via differentiation and ability to modulate the surrounding microenvironment through secretion of trophic factors [17,18,19]. They are characterized by their ability to adhere to plastic, self-renew, and differentiation potency into adipogenic, chondrogenic, and osteogenic cell types [20,21]. Human MSCs are commonly characterized by the expression, or lack thereof, of cell surface markers as CD73(+), CD90(+), CD105(+), CD106(+), CD44(+), CD45(−), CD31(−), CD34(−), and HLA-DR(−) [21,22,23]. However, MSCs are a heterogeneous lot both in terms of tissue of origin and within population; single cell sequencing and advances in multiparametric flow cytometry are defining further subpopulations. Moreover, the original locations must be taken in consideration when isolating MSCs. For example, bone marrow-derived MSCs (BM-MSCs) have positive biomarkers CD73, CD90, CD105, CD106, CD44, CD10, CD13, CD140β, CD146, and CD271 [24,25] but need to be lacking hematopoietic lineage markers such as CD45, CD34, CD14 or CD11b, CD79α or CD19, and HLA-DR, to distinguish from hematopoietic stem cells [21]. LNGFR and integrin alpha-1 have also been used to purify a relatively homogeneous population of MSCs from bone marrow. Murine BM-MSCs also share CD73(+), CD105(+), CD106(+), CD44(+), CD45(−), CD31(−), CD34(−), CD45(−), CD34(−), and CD11b(−) markers with humans, but also express Sca-1(+) and CD29 (+) as well [26,27]. In addition, other MSC tissue populations such as adipose-derived MSCs maintain the CD73(+), CD90(+), CD105(+), CD106(+), CD44(+), CD45(−), and CD31(−) expression phenotype; but are distinct from BM-MSCs by expressing the additional markers CD36(+), CD34(+), CD106(−), and CD146(−) [28,29,30], as shown in Figure 1.

## 3. Mutual Homing between Tumors and MSCs 

MSCs are the key players in tumor progression [31]. It has been found that MSCs derived from various tissues can home to tumor sites. Tumor cells and MSCs are able to migrate into each other’s niches, as shown in Figure 2. BM-MSCs contribute to the tumor stromal construction by differentiating into endothelial cells or myofibroblasts within and mostly surrounding the tumors in transgenic mouse models of dissemination [32,33]. Accumulating evidence suggests that MSCs can be attracted to tumor sites through multiple signals, including inflammatory cytokines, chemokines, and growth factors [34]. Those signal molecules can be generated by tumor cells themselves or by other cells in TME, and inflammatory cells in particular [35]. Inflammatory cytokines, for example, interleukin-1 beta (IL-1β) and tumor necrosis factor alpha (TNF-α) secreted by immune cells could be acting as chemoattractants for MSCs chemotaxis to tumor sites [36,37]. IL-6 is highly secreted by tumor cells under hypoxic conditions and serves to attract and activate MSCs [38]. Growth factors, such as transforming growth factor-beta (TGF-β), could also induce migration of BM-MSCs [39]. Chemokine CCL25 released by multiple myeloma cells attracts BM-MSCs’ move to the tumor location. CCL5 and CXCL12 also play important roles in MSCs’ migration to the osteosarcoma cell line Saos-2 [40]. CXCL16 is a ligand of CXCR6, and it is proved to be involved in the migration of MSCs towards prostate cancer [41]. 

Most mesenchymal tissues are not sites of tumor metastasis with the exception of the bone marrow (and brain), which is one of the main sites of dissemination and lethal growth. As resident cells in bone marrow, MSCs contribute to the recruitment of cancer cells to the bone or bone marrow. Several studies have found that attraction of tumor cells to the bone marrow is dependent on the dual interactions of SDF-1α and its receptor CXCR4 in both cancer cells and MSCs [33,42,43]. More recently, a study has established an inverse relationship between a subpopulation of MSCs and homing of cancer to bone. Such MSC populations express markers of endothelial cells (CD31, CD144, CD146) and pericytes (CD146, CD140b), in addition to MSC markers [44]. These studies suggest that it is specific subpopulations of seemingly homogenous MSCs contribute to the site selectivity of seeding. 

## 4. MSCs in Tumor Metastasis

Tumor dissemination from the prostate results from a metastatic cascade that involves a series of phenotypic switches. This progression is described in detail in other publications (examples of which include [1,4,8,45]). In brief, a subset of the carcinoma cells undergo a cancer-associated epithelial to mesenchymal transition (cEMT) to separate from the primary tumor and intravasate into circulatory conduits (lymph and blood vessels). At the other end of the circulatory conduits, the cancer cells are arrested in the capillaries, extravasate into the parenchyma and then undergo a phenotypic reversion back to a more epithelial state (cMET), at least transiently, as this not only enables survival in a hostile ectopic environment, but also puts the cells in a state of quiescent dormancy. After a variable period of dormancy, the carcinoma cells again shift towards mesenchymal to enable outgrowth as a lethal metastasis. It should be noted that these phenotypic switches are neither complete nor stringent, in that markers of both epithelial and mesenchymal phenotypes may be present or absent during any state [46]; rather the phenotype relates to the cell functioning and surface expression and ligandation of E-cadherin. While the general outline of this epithelial-mesenchymal plasticity during tumor progression and the functional implications thereof are well described, the TME signals that impart the switches are not understood, and serve as the basis for this review, as we propose that MSCs provide many of these triggers.

MSCs can enhance tumor metastasis by imparting a cEMT. Upon recruitment to the tumor sites, MSCs communicate with tumor cells in multiple ways: (1) direct contact; (2) paracrine secretome including extracellular vesicles; (3) modify other cell types in TME, e.g., immune cells; (4) differentiate into fibroblasts, and (5) fusion with tumor cells [47,48,49,50]. While most studies examine the interaction of MSCs with primary tumor sites, a recent study has shown that circulating MSCs preferentially home to micrometastases [51]. Still, the effects of the invading, and resident MSCs on the tumor cell behaviors may be similar for both the primary and micrometastatic nodules.

### 4.1. Soluble Factors from MSCs Contribute in Tumor EMT

Cancer-associated epithelial to mesenchymal transition (cEMT) and reverting transition (cMET) are required for escape from the primary site and successful metastatic seeding during metastases, respectively [1,2,3]. A plethora of studies have shown that MSCs drive tumor progression, via cEMT in particular, as well as create pre-metastatic niches as supportive microenvironments to aid circulating tumor cell in colonization of the target organ [52]; this is accomplished through the crosstalk between the MSCs and tumor cells, as shown in Figure 3. First, tumor cells can be modulated by MSC-derived signals, including cytokines, chemokines, growth factor, and exosomes. Second, tumor cells in turn affect MSC differentiation and proliferation. Hence, the consequent effects of MSCs in tumor progression depends on the phenotype, molecular signals, and cellular behavior of both tumor cells and MSCs.

In the tumor sites, the chemokines released by immune cells attract MSCs from bone marrow or surrounding tissues; this secretion is increased in the presence of inflammation and may represent the link between inflammation and cancer progression and death. In this case, not only are the MSCs attracted to the tumor, but are also activated to release signals that affect the tumor cells. The chemokine receptors CXCR2, CXCR3, and CXCR4, as well as their ligands, are found to be involved in this progression [49,53,54,55]. Inhibition of CXCR4 to block the crosstalk between MSCs and tumor cells suppressed hepatocellular carcinoma and osteosarcoma cell line proliferation, migration, and invasion. Inhibition of CXCR4 in the MSCs abolished elevated VEGF secretion, as well as p-Erk and p-Akt levels [53,55]. Increased CXCR2 and its ligands CXCL1, CXCL5, and CXCL7 were found in co-cultured MSCs and breast cancer, with this axis augmenting breast cancer cell migration [49,54]. Moreover, TNF-α stimulated MSCs released more CXCL9, CXCL10, and CXCL11, which are the ligands of CXCR3 [56], to increase cell migration of co-cultured breast cancer cell line MDA-MB-231 via the NF-κB signaling pathway [57].

Several cytokines including IL-6, IL-8, IL-10, and VEGF have been found to be secreted at higher levels by MSCs activated by IL-37, macrophages, or tumor cells [47,58,59,60]. Within the tumor microenvironment, IL-6 signaling is generally considered a malevolent player, promoting tumor progression [61]. IL-6 was firstly found mediating the communications between osteosarcoma cell line Saos-2 and MSCs; a manner in which proliferation was promoted by each other’s conditioned media [47]. MSCs isolated from colorectal tumors (CC-MSCs) secreted very high levels of IL-6 in vitro. With the presence of conditioned media from CC-MSCs, colorectal cancer cell lines SW48 and SW480 underwent cEMT, concomitant with enhanced cell migration, invasion, and proliferation [62]. Co-injecting CC-MSCs and SW48 cells in a murine xenograft model enhanced the tumor growth and lung metastasis. IL-6 in the CC-MSC-conditioned media activated Stat3 through Jak2, and PI3K-Akt signaling pathways [62], that latter also being a pathway that promotes tumor cell survival [6]. Macrophages, a major cell component in TME, are able to active MSCs with increased IL-6 and CXCL10 secretion as well [58,63].

Growth factors also play an important role in the interaction between tumor cells and MSCs. TGF-β family members drive cEMT [64]. IFN-γ and/or TNF-α primed adipose-derived MSCs (AD-MSCs) reduced E-cadherin expression in breast cancer cell line MCF-7 via elevated TGF-β1 signaling [65]. This is also true in the melanoma cell line B16. MSC-conditioned media mediated B16 cEMT via the TGF-β1/snail signaling pathway. Co-injection of MSCs with B16 promoted xenografted tumors cEMT [66]. VEGFC and fibroblast growth factor 10 (FGF10) may also participate in the process of tumor cell EMT [59]. MSCs enhanced VEGF expression in tumor cells, accompanied with the activation of RhoA-GTPase and ERK1/2, and a “reprogramming” of tumor growth [67]. Epidermal growth factor (EGF) from breast cancer-associated MSCs promote mammosphere formation via the PI3K/Akt signaling pathway [68].

### 4.2. MSCs Influence Tumor EMT via Multiple Signals

MSCs drive tumor progression not only through these soluble factors but also matrix changes [69,70,71]. When co-cultured BM-MSCs with colon cancer cell line KM12SM, the cEMT related matricellular factors—fibronectin, SPARC, and galectin 1—were only found elevated at the boundary of cancer clusters where direct contact happens [50]. The tumor-progression linked matricellular protein tenascin-C is found at the front of invading tumors and within metastatic tumors to promote survival [4,72]. While these matrix components can be generated by the tumor cells themselves or resident fibroblasts and macrophages, activated MSCs also produce these matricellular signalers at high levels [73,74,75]. Still, more extensive studies are needed to define the contribution of the MSCs in this mode of tumor progression.

Cell fusion may play a crucial role in cancer progression and leads to massive aberrant genotypic and phenotypic changes, though this is still controversial as to the extent in the human situation. It has been reported MSCs are able to fuse with breast, ovarian, lung, gastric, liver, and myeloma cell lines to modulate cancer cell proliferation and cEMT. Fusion of MSCs with SK-OV-3 cells—an ovarian cancer cell line—contributed to the generation of new cancer hybrid populations displaying a significantly reduced tumorigenicity [76]. However, most evidence has shown fusion of MSCs with cancer cells augmented its tumorigenic or metastatic abilities. Lewis lung cancer (LLC) or gastric cancer cell lines and MSC hybrid cells showed enhanced metastatic capacity and characteristics of cancer stem cells by undergoing cEMT. Instead of promoting gastric cancer cell proliferation, the cell cycle was blocked in the G0/G1 phase with elevated expression of p21, p27, and p53 in fused lung cancers [77,78]. This growth suppression in fusion progeny might be mediated by FOXF1 [79]. Fusion of rat BM-MSCs with human liver cancer cells HepG2 with low metastatic potential enhanced EMT promoting markers, such as vimentin, Twist, Snail, and matrix metalloproteinase (MMP) 2 and 9 activities. However, E-cadherin, the presence of which defines the epithelial phenotype, was increased [48]. Meanwhile, the fused cells generated increased numbers of metastatic liver and lung lesions. Moreover, myeloma cells and BM-MSC hybrid cells acquired more stemness potential and increased chemoresistance [80]. Still, whether these findings are generalizable to the human situation remains to be determined.

A specialized situation for promoting tumor progression and survival may occur through the well-described angiogenic actions of MSCs. After being co-cultured with AD-MSCs, the secretome of the breast cancer cell line MCF7, but not MDA-MB-231, was found to be more angiogenic; this was mediated by CXCL1 and CXCL8 released by AD-MSCs [81]. Similar results were also found in the prostate cancer cell line DU-145 [82]. To be notable, both MCF7 and DU-145 are E-cadherin positive cell lines, but MDA-MB-231 is E-cadherin negative; suggesting that this may be a switch to progress the non-metastatic epithelial cells towards dissemination. In mice, MSC-secreted IL-6 mediates cancer cells releasing proangiogenic factor endothelin-1 (ET-1) [83]. Angiogenic factors such as leukemia inhibitory factor (LIF), macrophage colony-stimulating factor (M-CSF), macrophage inflammatory protein-2 (MIP-2), and VEGF are highly generated by MSCs when co-cultured with melanoma B16 cells [84]. Moreover, MSC-derived IL-8 drives angiogenesis in colorectal carcinoma models [85]. Interestingly, aquaporin 1 (AQP1)—a water channel known to promote metastasis—was found to be increased significantly in osteosarcoma and hepatocellular carcinoma cells exposed to conditioned media from BM-MSCs [86]. These studies suggest MSCs may drive progression via upregulation of molecules already linked to metastasis.

### 4.3. MSCs in Tumor Chemoresistance

One of the most daunting aspects of metastatic tumors is the generalized resistance to a variety of therapies displayed by dormant and emergent metastases [4]. This is likely due to signals from the TME that imparts both cancer cell-intrinsic events (such as E-cadherin mediated survival during cMET [6]) and extrinsic signaling events. IL-6 secreted by MSCs not only regulates tumor cells’ cEMT, but also induces the chemoresistance. Exposure of Saos-2 and U2-OS cells to MSC-conditioned media increased the viable cells in the presence of therapeutic concentrations of doxorubicin or cisplatin [87]. Meanwhile, these pro-proliferation effects were accompanied by reduced caspase 3/7 activity and Annexin V binding. The activated IL6-STAT3 pathway increased expression of multidrug resistance protein (MRP) and P-glycoprotein (MDR-1) [87,88]. Interestingly, the secretion of IL-6 is also under the control of chemotherapy treatment. Part of MSCs entered a senescent phase rather than apoptosis with cisplatin pre-treatment, showed marked changes in phosphorylation profiles of many kinases, as well as increased secretion of IL-6 and IL-8. It led to increased chemoresistance and stemness of breast cancer cells [89]. Given the key role of IL-6 in MSC-associated tumor metastasis and chemoresistance, and being secreted by many types of cells, it is a very promising approach to metastatic cancer therapy by targeting IL-6 and its signaling pathways in TME. In addition, the Wnt-β-catenin pathway is activated in human cholangiocarcinoma cell line QBC939 by coculturing with umbilical cord-derived MSCs, promoting tumor cell proliferation, chemoresistance, and metastasis [90]. The ability of the MSCs to contribute to a bio-active matrix containing pro-survival matricellular components including tenascin-C [91,92] are additional ways that MSCs may contribute to generalized chemoresistance.

### 4.4. Immune Modulation of MSCs in TME

In the solid tumors, the heterogenic population of tumor cells collaborate with MSCs and immune cells, forming a “vicious triangle” of tumor development [93,94]. Besides communicating with tumor cells directly, MSCs modulate the majority of immune cells in TME, such as macrophages, neutrophils, and nature killer cells, to affect tumor progression. MSCs isolated from spontaneous lymphomas in mouse (L-MSCs) strikingly enhanced tumor growth in comparison to BM-MSCs, by recruiting monocytes/macrophages via the CCR2 signal [94]. Similarly, macrophage-activated MSCs acquired pro-inflammatory phenotype and promoted gastric cancer growth in an NF-κB dependent manner [95]. However, macrophages are heterogeneous in population and can be classified within a spectrum of M1 or M2, polarizing dependent on the stimuli present at time of activation. Tumor-associated macrophages (TAMs) isolated from solid and metastatic tumors have a wound suppressive M2-like phenotype; a phenotype that drives cEMT [96,97]. Interestingly, MSCs educated by M1 macrophage-conditioned media possessed a greatly enhanced ability in promoting tumor growth [98]. Such MSCs expressed high levels of iNOS and MCP1, which in turn increase TAM recruiting. Meanwhile, IL-6 secreted by M1-conditioned media educated MSCs that could polarize infiltrated TAMs into M2 like macrophages [98]. Pancreatic tumor-associated MSCs could recruit monocytes or macrophages and promote alternative macrophage polarization rather than the classical subtype as well [99]. Therefore, MSCs could contribute to tumor cell EMT indirectly via modulating macrophage polarization [97]. In a study regarding the MSCs and tumor progression, tumor cell-derived exosomes affected neither the growth factor production nor the immunosuppressive property of MSCs; rather, they endowed MSCs with a strong ability to promote macrophage infiltration into the tumor by producing plenty of CCR2 ligands, CCL2 and CCL7 [100].

Besides TAMs, tumor-associated neutrophils (TANs) in the TME contribute to tumor progression, invasion, and angiogenesis as well. TNF-α-activated MSCs strikingly enhanced tumor metastasis compared with normal MSCs. Various chemokines were increased in TNF-α-activated MSCs and among them, CXCR2 ligands (CXCL1, CXCL2, and CXCL5) efficiently recruited CXCR2+ neutrophils into the tumor and were responsible for its pro-metastatic effect [101]. An interesting study was performed to investigate the interaction of neutrophils and MSCs when primed by gastric cancer-derived MSCs (GC-MSCs). GC-MSCs activated neutrophils with increased expression of IL-8, TNF-α, CCL2, and oncostatin via STAT3 and ERK1/2 pathways, consequently augmented the migration of gastric cancer cells in a cell contact dependent manner. In turn, GC-MSC primed neutrophils induced the differentiation of normal MSCs into cancer-associated fibroblasts [102]. 

## 5. MSCs Differentiation in TME

MSCs have been investigated mainly as to their pro-tumor effects; but they may also contribute inhibitory stimuli as has been reviewed recently [31]. Multiple aspects regarding these contradictory functions of MSCs in tumors may be context dependent, with MSC heterogeneity being key. First, the origin of MSCs used in the studies might have the distinct characters that led to the differential functions. Second, intact or primary, tumor-derived or immune cell-primed MSCs have distinct secretomes. Lee et al. reported that TNF-α pre-activated MSCs had anti-tumor activity secondary to expressing cell death inducing cytokine TRAIL in a TLR3-dependent manner [103]. Interestingly, prior to this finding and contradictorily, Waterman et al. proposed MSC polarizing resulted from TLR signaling—TLR4 stimulation promoted a pro-inflammatory MSC1, which attenuated tumor growth; whereas TLR3 promoted immunosuppressive phenotype MSC2 to polarize and promote tumor growth and metastasis [104,105]. The effects of MSCs on polarizing macrophages along the lines of pro-progression M2 or quiescing M1 spectrum phenotypes [97] is discussed above. 

Cancer-associated fibroblasts (CAFs) are a heterogeneous population of stromal cells in the microenvironment of solid tumors. CAFs are the most prominent stromal cell type, and their abundance was shown to correlate with worse outcomes. Nevertheless, the origin and function of CAFs in tumors are still unclear. However, emerging evidence has shown that MSCs can be a major source of CAFs. MSCs are able to be differentiated into fibroblast-like cells in vitro with breast cancer-conditioned media [106]. Notably, a relatively long term (30 days) incubation was necessary for the transition, indicating MSCs involved in tumor progression as CAFs is a late event after MSC invasion. Moreover, MSCs and CAF share many similarities, including surface markers, as well as capacity to differentiate to osteocytes, chondrocytes, and adipocytes [107]. These differentiation shifts lead to the difficulty in characterizing MSCs and CAF in patient tissues. Most recently, Raz et al. identified PDGFRα as being absent in bone marrow-derived CAFs, but present in resident CAFs. More importantly, BM-derived CAF recruiting decreased the percentage of PDGFRα positive CAFs, and decreased PDGFRα in breast cancer patients was associated with worse prognosis. It suggests that BM-derived CAFs may have more deleterious effects in tumor progression than resident CAFs [108]. 

## 6. Conclusions

The roles of MSCs in tumor growth and progression are only beginning to be discerned. From animal models, copious evidence points toward active attraction of MSCs to tumor sites along with deleterious results of promoting both dissemination, metastasis, and chemoresistance. However, these findings have not been supported in human specimens due to the potentially transient nature of these effects and the differentiation of the MSCs over time. Further, as MSC biology is complex with different subpopulations potentially pulling in opposite directions, further study is critical. Major questions that remain include (a) relative contribution of the circulating versus resident MSCs, (b) effects on the inflammatory network as to indirect promotion of progression including macrophage polarization, (c) effects on immune networks including impact on immunotherapies, (d) differentiation into components of the tumor organ including CAFs, and (e) roles in dormancy and emergence both directly and via altering the TME. 

## Figures and Tables

**Figure 1 ijms-20-02566-f001:**
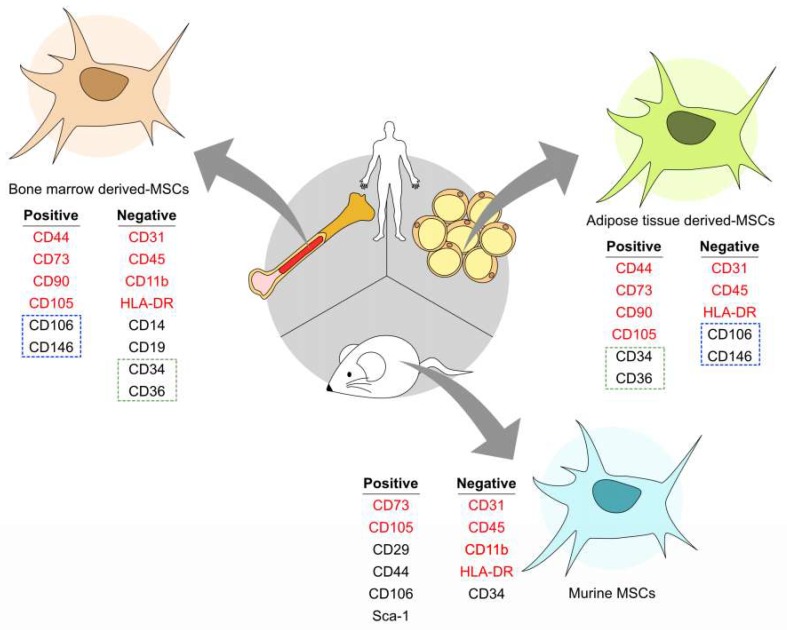
Cell surface markers of vary between mesenchymal stem cells/multipotent stromal cells (MSCs). Human bone marrow-derived (BM)-MSCs share most of the markers such as CD44, CD73, CD90, and CD105 with adipose-derived (AD)-MSCs. CD106 and CD146 (outlined with the blue dotted line) are positive in BM-MSCs, but negative in AD-MSCs. Both are negative with CD31, CD45, and HLA-DR, whereas BM-MSCs are negative with CD34 and CD36 (outlined with the green dotted line), which are positive in AD-MSCs. Mouse MSCs express Sca-1 specifically. The image was generated with Affinity Designer 1.6.1.

**Figure 2 ijms-20-02566-f002:**
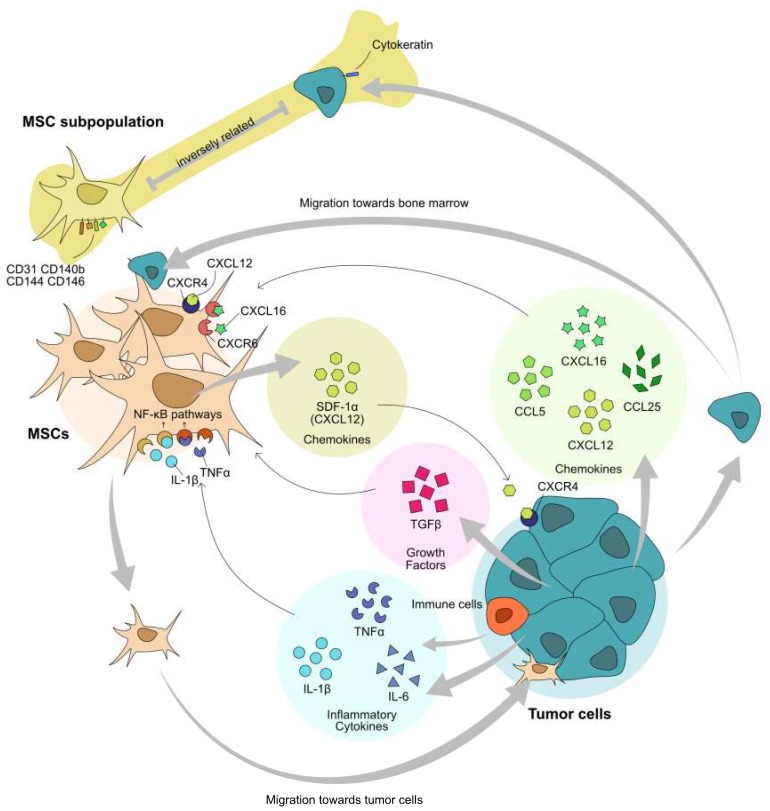
MSCs and tumor cells migrate towards each other. Circulating bone marrow or adipose tissue resident MSCs are recruiting to the tumor sites in the direction of multiple factors released by tumor or immune cells in the tumor microenvironment (TME). In turn, BM-MSCs could attract tumor cells into the bone morrow mediated by chemokines and their receptors. However, the subpopulation of MSCs in bone marrow inversely correlates with the cytokeratin+ cell numbers in prostate and breast cancer. Bold and thin arrows refer to cellular and molecular communications, respectively. The image was generated with Affinity Designer 1.6.1.

**Figure 3 ijms-20-02566-f003:**
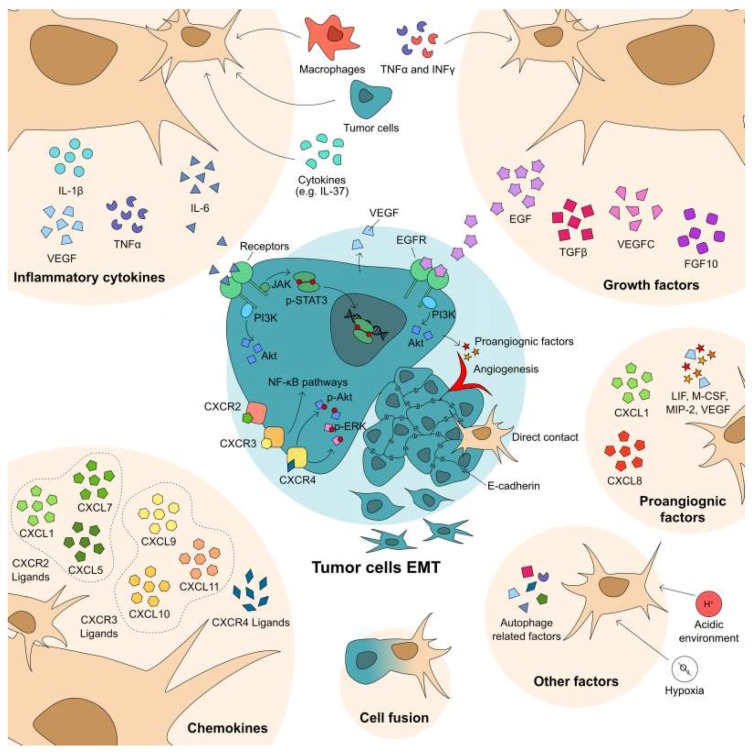
MSCs regulate tumor cell cancer-associated epithelial to mesenchymal transition (cEMT) in multiple levels. MSCs promote tumor cells via direct contact, or secretome, including cytokines, growth factors, and chemokines. Also, MSCs and tumor hybrid cells acquire mesenchymal or stemness. Moreover, many other factors, such as angiogenesis, autophagy, acidic, or hypoxia, contribute to tumor cells’ cEMT. The image was generated with Affinity Designer 1.6.1.

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
