# Peer review of "Adult Stem Cell Functioning in the Tumor Micro-Environment"

_ijms, 2019, doi:10.3390/ijms20102566_

Round 1
Reviewer 1 Report
In the current review paper, Yuhan et al. set up to summarize functions of adult stem cells in tumor microenvironment. Although this is certainly a very interesting topic for readership working in cancer research, this very broad and always recurring theme is difficult to approach efficiently within a single review manuscript. As such, the current paper presents with particular strengths (nice and informative figures) but on the other hand has several weaknesses that should be addressed prior potential publication.
Specific comments:
1. Due to the broadness of the topic, the manuscript lacks a clear point and seems to be a bit unfocused. It does not read very easily and fails to pass to the readers a meaningful information. It would be of a great value if some basic questions could be answered between the lines by the authors: What is the purpose of this review? What the readers should expect and what messages the authors wish to deliver in particular?
2. Although in section 1.1 the authors are somehow characterizing the cellular population they wish to address, a more focused description of these cells (e.g., their peculiarities in comparison to stem cells in general, their niches, etc.) should be clearly but concisely highlighted.
3. The structure of the manuscript could be improved – what is the point to have a subsection 1.1 within the Introduction if not other subsections (1.2, 1.3 or so) appear there? Also, in relation to other subchapters, section 3.1 is very lengthy etc. It would be useful if the structure of the manuscript would be reworked.
4. The current version of the manuscript contains many grammatical errors and spelling mistakes, which makes the text difficult to follow. A thorough editing by a native speaker would be necessary.
5. Some statements seems to be lacking proper referencing – if an important information is transmitted from the same paper in longer stretches of text, it would be worthwhile to repeat the reference in the brackets (e.g., p.6, line 143, p7., line 217 etc.).
6. P.2, line 47 – what role has been proposed for tissue and circulating stem cells? Could you elaborate here?
The figures are very nice. However, their legends do not appear to be sufficiently clear – for example, why both red and black letters as well as blue squares are used for markers in Figure 1?
Author Response
Reviewer 1:
In the current review paper, Yuhan et al. set up to summarize functions of adult stem cells in tumor microenvironment. Although this is certainly a very interesting topic for readership working in cancer research, this very broad and always recurring theme is difficult to approach efficiently within a single review manuscript. As such, the current paper presents with particular strengths (nice and informative figures) but on the other hand has several weaknesses that should be addressed prior potential publication.
Specific comments:
1. Due to the broadness of the topic, the manuscript lacks a clear point and seems to be a bit unfocused. It does not read very easily and fails to pass to the readers a meaningful information. It would be of a great value if some basic questions could be answered between the lines by the authors: What is the purpose of this review? What the readers should expect and what messages the authors wish to deliver in particular?
Response 1: We appreciate this insightful comment and have amended the text to direct the reader. In the “Abstract” part, we mention “Herein, we review emerging evidence as to how these cells, both from distant sources, mainly the bone marrow, or local resident cells, can impact the progression of solid tumors. These nascent investigations raise more questions than they answer, but paint a picture of an orchestrated web of signals and interactions that can be modulated to impact tumor progression.” This review aims to highlight the role of MSCs in tumor progression, metastasis in particular, in order to provide the advanced and comprehensive information on interaction of MSCs and tumor cells in both primary and metastatic sites. To make it even more clear, related statements are added to the Introduction and throughout.
2. Although in section 1.1 the authors are somehow characterizing the cellular population they wish to address, a more focused description of these cells (e.g., their peculiarities in comparison to stem cells in general, their niches, etc.) should be clearly but concisely highlighted.
Response 2: To correct this omission, a brief explanation of MSCs function, location and characterization is now adding in the beginning of section 2.
3. The structure of the manuscript could be improved – what is the point to have a subsection 1.1 within the Introduction if not other subsections (1.2, 1.3 or so) appear there? Also, in relation to other subchapters, section 3.1 is very lengthy etc. It would be useful if the structure of the manuscript would be reworked.
Response 3: We apologize for confused categories. We have reorganized the numbering. Subsection “1.1 Human mesenchymal stem cells/multipotent stromal cells (MSCs)” is now an individual section “2. Characterization of human mesenchymal stem cells/multipotent stromal cells (MSCs)”. Since subsection 3.1 is too long and hard to follow, we divide it to two parts, “4.1. Soluble factors from MSCs contribute in tumor EMT” and “4.2 MSCs influence tumor EMT via other ways”.
4. The current version of the manuscript contains many grammatical errors and spelling mistakes, which makes the text difficult to follow. A thorough editing by a native speaker would be necessary.
Response 4: We apologize. The typos and grammatical errors have been corrected and tracked. All authors (include 3 native English speakers) have read this through again.
5. Some statements seems to be lacking proper referencing – if an important information is transmitted from the same paper in longer stretches of text, it would be worthwhile to repeat the reference in the brackets (e.g., p.6, line 143, p7., line 217 etc.).
Response 5: Detailed citations are now included.
6. P.2, line 47 – what role has been proposed for tissue and circulating stem cells? Could you elaborate here?
Response 6: We apologize for this confusing sentence and have clarified it to define that both resident and recruited stem cells are operating in the TME.
7. The figures are very nice. However, their legends do not appear to be sufficiently clear – for example, why both red and black letters as well as blue squares are used for markers in Figure 1?
Response 7: We apologize for omitting the information. The figure legends are now expanded.
Reviewer 2 Report
Yuhan et al. presented a very complete and extensive review concerning the role of mesenchymal stem cells/multipotent stromal cells (MSCs) in correlation to Tumor Micro-Environment (TME), and its function in tumor progression and metastasis.
Authors touched all aspects and fields in this topic, also the contradditory one, and this reviewer requires only minor revisions, as stated below.
1) Only a punctualization: this referee suggests to add a very brief explanation of the epithelial to mesenchymal transition (EMT) and the cancer-related EMT, to better comprehend the importance of MSCs in this event.
2) The manuscript should be carefully edited, because there are frequent typos throughout the manuscript (for example, page 1, line 24, “matricellular proteins” have been inserted twice).
Author Response
Reviewer 2:
Yuhan et al. presented a very complete and extensive review concerning the role of mesenchymal stem cells/multipotent stromal cells (MSCs) in correlation to Tumor Micro-Environment (TME), and its function in tumor progression and metastasis.
Response 1: We appreciate this comment, especially the first author for whom this is his first paper.
Authors touched all aspects and fields in this topic, also the contradictory one, and this reviewer requires only minor revisions, as stated below.
1) Only a punctualization: this referee suggests to add a very brief explanation of the epithelial to mesenchymal transition (EMT) and the cancer-related EMT, to better comprehend the importance of MSCs in this event.
Response 1: Thanks for suggesting such a good point. A short paragraph is now included in section 4 to briefly describe developmental EMT and cancer-related EMT.
2) The manuscript should be carefully edited, because there are frequent typos throughout the manuscript (for example, page 1, line 24, “matricellular proteins” have been inserted twice).
Response 2: We apologize and have corrected these typos and mistakes.
Round 2
Reviewer 1 Report
The authors have addressed all the concerns in a satisfactory manner.